# Identification of miRNAs in Response to Sweet Potato Weevil (*Cylas formicarius*) Infection by sRNA Sequencing

**DOI:** 10.3390/genes13060981

**Published:** 2022-05-30

**Authors:** Jian Lei, Yuqin Mei, Xiaojie Jin, Yi Liu, Lianjun Wang, Shasha Chai, Xianliang Cheng, Xinsun Yang

**Affiliations:** 1Institute of Food Crops, Hubei Academy of Agricultural Sciences, Wuhan 430064, China; leijian2006@hbaas.com (J.L.); 20095131210148@hainanu.edu.cn (Y.M.); xiaojiejin@hbaas.com (X.J.); 201973042@yangtzeu.edu (Y.L.); wanglianjun@hbaas.com (L.W.); chaishasha2008@hbaas.com (S.C.); chengxianliang@hbaas.com (X.C.); 2College of Horticulture, Hainan University, Haikou 570228, China; 3Yangtze University, Jingzhou 434022, China

**Keywords:** sweet potato weevil, microRNA, sRNA sequencing, target gene

## Abstract

The sweet potato weevil (*Cylas formicarius*) is an important pest in the growing and storage of sweet potatoes. It is a common pest in the sweet potato production areas of southern China, causing serious harm to the development of the sweet potato industry. For the existing cultivars in China and abroad, there is no sweet potato variety with complete resistance to the sweet potato weevil. Thus, understanding the regulation mechanisms of sweet potato weevil resistance is the prerequisite for cultivating sweet potato varieties that are resistant to the sweet potato weevil. However, very little progress has been made in this field. In this study, we inoculated adult sweet potato weevils into sweet potato tubers. The infected sweet potato tubers were collected at 0, 24, 48, and 72 h. Then, a miRNA library was constructed for Eshu 6 and Guang 87 sweet potato tubers infected for different lengths of time. A total of 407 known miRNAs and 298 novel miRNAs were identified. A total of 174 differentially expressed miRNAs were screened out from the known miRNAs, and 247 differentially expressed miRNAs were screened out from the new miRNAs. Moreover, the targets of the differentially expressed miRNAs were predicted and their network was further investigated through GO analysis and KEGG analysis using our previous transcriptome data. More importantly, we screened 15 miRNAs and their target genes for qRT-PCR verification to confirm the reliability of the high-throughput sequencing data, which indicated that these miRNAs were detected and most of the expression results were consistent with the sequencing results. These results provide theoretical and data-based resources for the identification of miRNAs in response to sweet potato weevil infection and an analysis of the molecular regulatory mechanisms involved in insect resistance.

## 1. Introduction

MicroRNAs are a group of short-sequence RNAs of non-coding proteins with a length of 20–24 nt which are widely distributed in eukaryotes and have high conservation, timing, and tissue specificity [1,2,3,4,5,6]. These miRNAs are not only involved in plant growth [7,8,9,10,11] and physiological metabolism [9,12,13], but also play a key regulatory role in various biotic and abiotic stress responses [14,15,16,17]. Studies have found that miRNAs play an indispensable role in plant defense against insects. Research on tobacco–TBM interactions has shown that changes to a plant’s secondary metabolites are induced by miRNAs after an insect infection to achieve insect resistance [18]. Before and after the TBM infection in wild-type tobacco and RDR1 tobacco plants, it was found that the RDR1 tobacco mutant was susceptible to some miRNAs that induced the expression of ethylene and jasmonic acid, signaling pathway-related genes and causing plant hormone changes. The phenotype of RDR1 was indirectly regulated by smRNAs due to the changes in hormone-signaling-pathway-related genes [19].

The sweet potato weevil (Fabricius), also known as the sweet potato ant elephant and the sweet potato elephant snout, is an important pest in the growing and storage of sweet potatoes [20]. It is a common pest in the sweet potato production areas of southern China, causing serious harm to the development of the sweet potato industry. To date, the research on sweet potato weevils in China has mainly focused on its biology and control [21,22,23,24], and there are few reports on the resistance gene resources for the sweet potato weevil [25,26]. For the existing cultivars in China and abroad, there is no sweet potato variety with complete resistance to the sweet potato weevil [27]. Therefore, understanding the regulation mechanisms of sweet potato weevil resistance is the prerequisite for cultivating sweet potato varieties that are resistant to the sweet potato weevil.

To date, research indicates that miRNAs are mainly involved in the regulation of growth, development, and stress responses in sweet potato [28,29]. Sun et al. [30] explored the expression of 16 miRNAs in different tissues of sweet potato plants and found that some miRNAs were expressed in an organ-dependent manner. The expression level of miR167 in stamens was higher than that in other tissues, indicating that it may be crucial for stamen development. The expression levels of miR156 and miR162 in the roots were significantly lower than those in the leaves and fibrous roots, indicating that these miRNAs may play a role in the initiation and development of roots. The inhibitory expression of miR408 enhanced the defense system of transgenic sweet potato plants against herbivore injury by up-regulating the expression levels of IbKCS, IbPCL, and IbGAUT [31]. However, the key miRNAs and their functions in the regulation of the sweet potato weevil have not been identified and analyzed, and the molecular mechanism of miRNAs in the regulation of sweet potato weevil infection is still unclear. 

In this study, high-throughput sequencing technology was used to study changes in the expression of miRNAs in sweet potatoes at the whole-genome level after an infection by sweet potato weevils; identify known miRNAs and new miRNAs; screen and excavate differentially expressed miRNAs in response to sweet potato weevils; and annotate and enrich the predicted regulatory target genes. This research provides theoretical and data-based references for further exploring the regulatory mechanisms of miRNAs and their molecular regulatory pathways and networks in the response and adaptation to sweet potato weevil infection.

## 2. Materials and Methods

### 2.1. Plant Materials

The sweet-potato-weevil-susceptible sweet potato variety Eshu 6 and the sweet-potato-weevil-resistant sweet potato variety Guang 87 were selected as experimental materials. The two varieties were planted in 50 pots each and placed in the potted plant section at the Institute of Food Crops, Hubei Academy of Agricultural Sciences.

### 2.2. Infection Treatment and Sample Collection

Sweet potato plants with a strong growth potential and relatively consistent growth were selected and placed in insect cages, with 1 pot per cage and 3 pots per variety. The environmental conditions were 25 ± 2 °C, 80% relative humidity, 16 h of light, and 8 h of darkness. Approximately 20 adult sweet potato weevils were inoculated to infect sweet potato tubers. The infected sweet potato tubers were collected after 0, 24, 48, and 72 h and named E_0, E_24, E_48, E_72, G_0, G_24, G_48, and G_72, respectively. Each experimental group contained three biological replicates, for a total of 24 samples (2 treatments × 4 time points × 3 biological replicates). The samples were promptly frozen in liquid nitrogen and stored at −80 °C until required.

### 2.3. Library Preparation and Small RNA Sequencing

The RNA of the 24 infected sweet potato tuber samples was extracted and used for sRNA library construction. The libraries were sequenced on an Illumina Noveseq platform at Wuhan Feisha Gene Information Co., Ltd. (Wuhan, China).

### 2.4. Identification of miRNA

Sequences were mapped to the reference genome (http://public-genomes-ngs.molgen.mpg.de/SweetPotato/ (accessed on 20 March 2021)) using bowtie with the parameters (-p 5 -v 1 -k 1) and assessed for the mapping rate and genome distribution. The mapped sequence reads were compared to known sequences in miRBase (v. 22) using miRDeep2 (-g 0) and sRNA-tools-cli (tool hp_tool) [32,33] to identify known miRNAs. The miREvo [34] and miRDeep [35] software was used to predict the candidate miRNAs and assess length distribution and nucleotide proportion to identify novel miRNAs.

### 2.5. Identification of Differentially Expressed miRNA

The expression levels of miRNAs in all samples were standardized or normalized by the TPM value. Eshu 6 and Guang 87 samples at 24 h, 48 h, and 72 h after treatment (E/G_24, E/G_48 and E/G_72) were compared with the corresponding samples at 0 h (E/G_0). DESeq2 [36] was used to identify the differently expressed miRNAs. The miRNAs with |log2(fold change)| > 1 and an adjusted *p*-value < 0.05 were considered differentially expressed miRNAs.

### 2.6. Prediction of miRNA Targets

TargetFinder [37] and qTar were used to predict the target genes of differentially expressed miRNAs. GO enrichment analysis and KEGG enrichment analysis were performed on the candidate targets. GOseq [38] and KOBAS [39] were used for GO and KEGG enrichment analyses of the target genes.

### 2.7. qRT-PCR Analysis

Fifteen differentially expressed miRNAs and their target genes were selected. Specifically designed forward primers and universal reverse primers were used for mature miRNAs (Appendix A). Then, SYBR^®^ Select Master Mix (2X) was used for RT-PCR amplification. U6 (U6-F: GGGGACATCCGATAAAATT, U6-R: TGTGCGTGTCATCCTTGC) was used as the internal reference gene for the miRNA, and sweet potato β-Actin (β-Actin-F: AGCAGCATGAAGATTAAGGTTGTAGCAC, β-Actin-R: TGGAAAATTAGAAGCACTTCCTGTGAAC) was used as the internal reference gene for the target gene. Each sample was repeated three times. The relative expression of genes was calculated using the 2^−ΔΔCt^ method [40].

## 3. Results

### 3.1. High-Throughput Sequencing Data Analysis

A total of 24 samples were taken for sRNA library construction and sRNA sequencing. As shown in Table 1, 14,507,166, 14,810,952, 13,701,096, and 15,575,806 total sequenced reads were extracted from the infected Eshu 6 tubers at 0 h, 24 h, 48 h, and 72 h, respectively, while a total of 14,162,156, 15,108,182, 13,616,807 and 15,348,571 total sequenced reads were extracted from the infected Guang 87 tubers at 0 h, 24 h, 48 h, and 72 h, respectively. After filtering and quality control, 10,978,936, 13,084,249, 9,258,096, and 13,362,839 clean reads from Eshu 6 tubers and 11,968,368, 13,596,968, 11,742,576, and 14,047,853 clean reads from Guang 87 tubers were obtained and blasted to the reference sequences. An amount of 75.68–85.79% of clean reads for Eshu 6 samples and 84.51–91.53% of clean reads for Guang 87 samples were assigned to the reference genome (Table 1).

We counted the total clean reads and found that the length distribution patterns of the sRNAs were similar in the two libraries. The lengths ranged from 18 to 30 nt, of which 22 nt and 24 nt sRNAs were the most abundant (Figure 1). This result is consistent with previous reports by Tang et al. [41].

### 3.2. Identification of Known miRNAs and Novel miRNAs

By comparing with the miRbase database, a total of 407 known mature miRNAs and 908 known hairpin miRNAs were identified. A total of 298 novel mature miRNAs and 307 novel hairpin miRNAs were identified in the libraries (Table 2). These miRNAs were then divided into 83 identified families (Figure 2). Among them, the miR159 family had the largest number, with 56 members, followed by miR156 (51), miR166 (47), miR171 1 (40), miR395 (39), and miR167 1 (37).

### 3.3. Analysis of Differentially Expressed miRNAs

During the detection of differentially expressed miRNAs, |log2 (FC)| > 1 and *p* < 0.05 were used as the screening criteria. There were 421 differentially expressed miRNAs observed during the infection by sweet potato weevils, including 174 known miRNAs and 247 new miRNAs (Appendix A). 

The results (Table 3) show that, among the different treatment lengths within the same variety, the differential expression of miRNAs in Eshu 6 was the highest after 24 h of infection, while the differential expression of miRNAs in Guang 87 was the lowest after 24 h of infection. The number of up-regulated miRNAs was higher than down-regulated miRNAs. When comparing the treatments of different varieties for the same lengths of time, the differentially expressed miRNAs were the highest after 24 h of infection, and the differentially expressed miRNAs were the lowest after 48 h of infection.

In Figure 3a, 150 miRNAs were differentially expressed in Eshu 6 tubers at all three time points after treatment. The 150 differentially expressed miRNAs were divided into two categories. Class I contained 67 miRNAs, and their expression levels were down-regulated at 24 h, 48 h, and 72 h after infection. Class II contained 83 miRNAs which were up-regulated at 24 h, 48 h, and 72 h after infection. In Figure 3b, 107 miRNAs were differentially expressed in Guang 87 tubers at all three time points after treatment. These 107 differentially expressed miRNAs were divided into two categories. Class I contained 47 miRNAs, and their expression levels were down-regulated at 24 h, 48 h, and 72 h after infection. Class II contained 60 miRNAs, and their expression levels were up-regulated at 24 h, 48 h, and 72 h after infection. In Figure 3c, 26 miRNAs were differentially expressed in both Eshu 6 and Guang 87 tubers at all three time points after treatment. These 26 differentially expressed miRNAs were divided into two categories. Class I contained 5 miRNAs, and their expression levels were down-regulated. Class II contained 14 miRNAs, and their expression levels were up-regulated. The results show that the expression levels of the same miRNAs in both the same and different varieties, and after infection for different lengths of time, show the same trend in variation.

### 3.4. Predicted miRNA Target Genes and GO/KEGG Enrichment Analyses

Target genes were predicted using the TargetFinder and qTar software according to the sequence information for the differentially expressed miRNAs and the corresponding species. A total of 33,909 identical potential target genes were predicted by the two prediction methods, of which 21,717 were known miRNA target genes and 12,192 were new miRNA target genes (Appendix A).

To further understand the metabolic pathways and biological processes of differentially expressed miRNAs after sweet potato weevil infection, GO and KEGG enrichment analyses were performed. GO analysis was performed on the predicted target genes and identified three main functional categories in the response of sweet potato to sweet potato weevil infection: molecular function, cellular components, and biological processes (Appendix A). In the biological process category, the main biological functions of the predicted target genes are enriched in metabolic and cellular processes. In the cellular component category, the predicted biological functions of target genes are mainly enriched in the cell, organelles, and membrane. In the molecular function category, the predicted biological functions of target genes were mainly enriched in binding, catalytic activity, and transporter activity. Most target gene functions are related to these and other similar binding functions. KEGG analysis showed that the target cells were enriched in the MAPK signaling pathway–plant, phosphatidylinositol production, plant hormone production, ascorbate production, aldarate metabolism, α-linolenic acid metabolism, etc. In conclusion, these results provide clues and references for revealing the molecular and cellular mechanisms involved in the response of sweet potato plants to infection by sweet potato weevils.

The target genes regulated by these miRNAs play a key regulatory role in the insect resistance of sweet potato (Table 4). The results show that three members of the MIR167_1 family (bna-miR167d, vvi-miR167c, and ptc-miR167f-5p), one member of the MIR156 family (hbr-miR156), and one member of the n_MIR318 family (novel_318) all possess a target gene related to insect resistance. Among them (Appendix A), the expression level of bna-miR167d was up-regulated at 24 h and 48 h after sweet potato weevil infection of Eshu 6 tubers; the expression levels of vvi-miR167c and ptc-miR167f-5p were up-regulated at 24 h and 48 h after sweet potato weevil infection of Eshu 6 tubers and 48 h after sweet potato weevil infection of Guang 87 tubers; and the expression level of hbr-miR156 was up-regulated at 24 h after sweet potato weevil infection of Eshu 6 tubers and 72 h after sweet potato weevil infection of Guang 87 tubers. The expression level of novel_318 was down-regulated after Eshu 6 was infected by sweet potato weevil for 72 h, but up-regulated at 0 h, 24 h, and 72 h after the infection of two varieties with different insect tolerances for the same length of time. The results show that these five miRNAs were the key regulatory factors in the response of sweet potato plants to the infection mechanism of sweet potato weevils.

### 3.5. qRT-PCR Verification

A total of 15 differentially expressed miRNAs and their target genes, including ath-miR319a, gma-miR168b, cpa-miR166e, mtr-miR319a-3p, zma-miR166h-3p, ath-miR396a-3p, osa-miR166g-3p, ath-miR166a-3p, lja-miR166-3p, novel_47, aau-miR168, osa-miR166d-5p, gma-miR396a-3p, ath-miR168a-3p, and novel_136, were screened for RT-PCR verification in the non-infected and infected treatments at different stages.

According to the results in Figure 4, the expression patterns of the 15 selected miRNAs in qRT-PCR experiments were consistent with those detected by high-throughput sequencing. Similar expression trends (up-regulation or down-regulation) were observed between the qRT-PCR analysis and the sRNA sequencing results.

## 4. Discussion

When induced by insect stress, miRNAs change their expression level and participate in plant–insect defense responses by regulating the expression of stress-response target genes at transcriptional and post-transcriptional levels [42,43,44]. As discussed, studies have found that miRNAs play an indispensable role in plant defense against insects. However, there have been relatively few studies conducted on miRNAs in sweet potato, and studies to date have mainly focused on the miRNAs related to growth, development, and the regulation of stress responses [45,46,47]. Studies on the miRNAs involved in sweet potato resistance to insect stress are also rarely reported. In this study, we analyzed the changes in the miRNA transcriptome for the whole sweet potato genome in order to uncover the miRNAs related to sweet potato weevil infection, and we also analyzed the sweet potato’s regulatory network in response to sweet potato weevil stress.

Under insect stress, miR156 is up-regulated, which controls the synthesis of plant secondary metabolites and the formation of epidermal hairs. It also helps the plant to resist insect infection by controlling the expression of target genes encoding SPL-family transcription factors [48,49,50]. It has been reported that miR156-SPL14 affects the JA and JA-Ile (jasmonoyl-isoleucine) content by regulating the expression of MPK6 and other genes in the JA pathway, thereby reducing the fecundity and survival rate of BPH [51]. The target gene of PhmiR156 in *Phyllostachys pubescens*, PhSPL17, plays an important role in managing the insect resistance of *Phyllostachys pubescens* [52]. A study on the regulation mechanisms of epidermal hair distribution in A. *thaliana* showed that SPL9 can directly bind to the promoter on the negative regulator gene TCL1. In the growth of plants, miR156 levels decreased, SPL9 increased gradually, and the TCL1 gene expression levels increased, thereby inhibiting the formation of epidermal hairs on the inflorescence axis and the floral organs [53]. The results of this study show that the expression level of hbr-miR156 was up-regulated after the sweet potato tuber was infected with sweet potato weevils, and the predicted target gene hbr-miR156 (Tai6.44728.1) may have a negative regulatory relationship with hbr-miR156 as suggested by qRT-PCR verification. However, this molecular regulation mechanism needs to be further studied in the future.

At present, the research on miR167 is focused on Arabidopsis thaliana, rice, tomato, and other model plants, and most of the research focuses on the regulation of plant growth and development. In sweet potato, only the expression of miR167 in different tissues was detected. It was found that the expression level of miR167 in sweet potato stamens was higher than that in other tissues, indicating that miR167 may be essential for stamen development. However, there is no report on the role of miR167 in the regulation of stress responses to sweet potato diseases and insect pests. In this study, bna-miR167d, vvi-miR167c, and ptc-miR167f-5p (members of the three miR167 families) were significantly up-regulated by sweet potato weevil infection. The corresponding predicted target genes may also have a negative regulatory relationship as verified by qRT-PCR. Therefore, further studies on the molecular regulation mechanism of miR167 are needed. Moreover, many of the new differentially expressed miRNAs found in this study may be closely related to the regulation of sweet potato weevil infection stress, and their specific molecular mechanisms need to be further explored and analyzed.

When plants are infected by pests and diseases, insect-resistant defense mechanisms are initiated at the molecular level through insect-responsive miRNAs and their target genes, which control the synthesis of plant secondary metabolites and the formation of epidermal hairs to resist insect infection [18,54,55]. In this study, the target genes corresponding to differential miRNAs were found to be significantly enriched in the MAPK signaling pathway–plant, phosphatidylinositol production, plant hormone production, ascorbate production, aldarate metabolism, α-linolenic acid metabolism, etc., confirming that these biological processes and metabolic pathways may be involved in the infection response to the sweet potato weevil. However, their specific mechanisms remain to be further explored. 

In conclusion, miRNAs play an important role in the adaptation and response process of sweet potato plants to the infection stress caused by the sweet potato weevil. The regulation mechanisms of miRNAs and their target genes analyzed in this paper provide a theoretical basis for analyzing the response of sweet potato to biological stressors, and provide molecular data resources for genetic breeding to improve the insect resistance of sweet potato.

## Figures and Tables

**Figure 1 genes-13-00981-f001:**
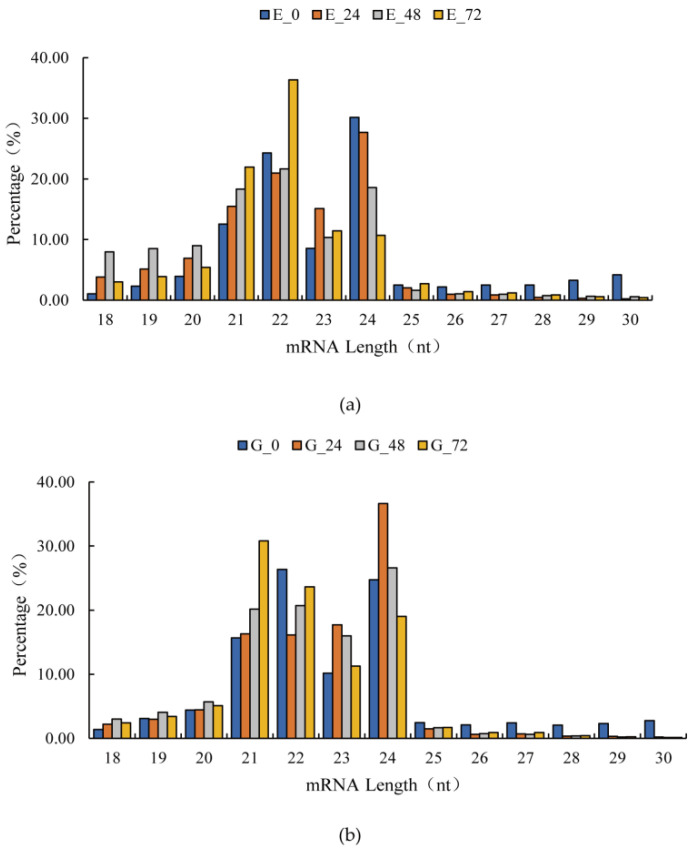
The sRNA length distribution. (**a**) Sequencing frequency of miRNAs in Eshu 6 with different lengths; E_0, E_24, E_48, and E_72: the infected Eshu 6 tubers at 0 h, 24 h, 48 h, and 72 h. (**b**) Sequencing frequency of miRNAs in Guang 87 with different lengths; G_0, G_24, G_48, and G_72: the infected Guang 87 tubers at 0 h, 24 h, 48 h, and 72 h.

**Figure 2 genes-13-00981-f002:**
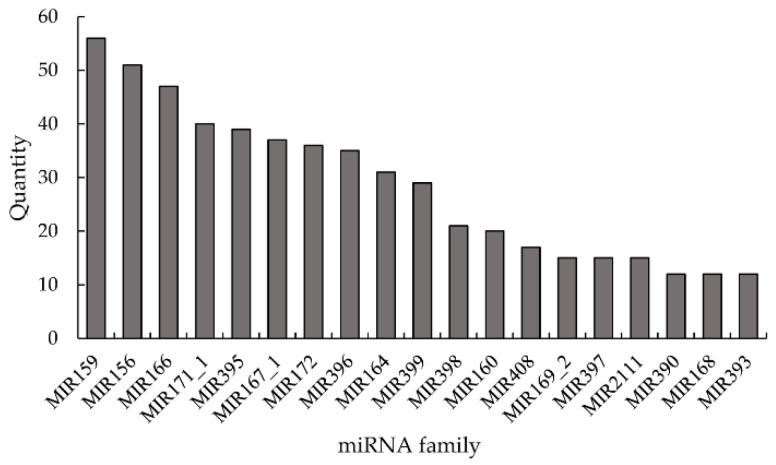
The number of miRNAs belonging to the top 19 miRNA families.

**Figure 3 genes-13-00981-f003:**
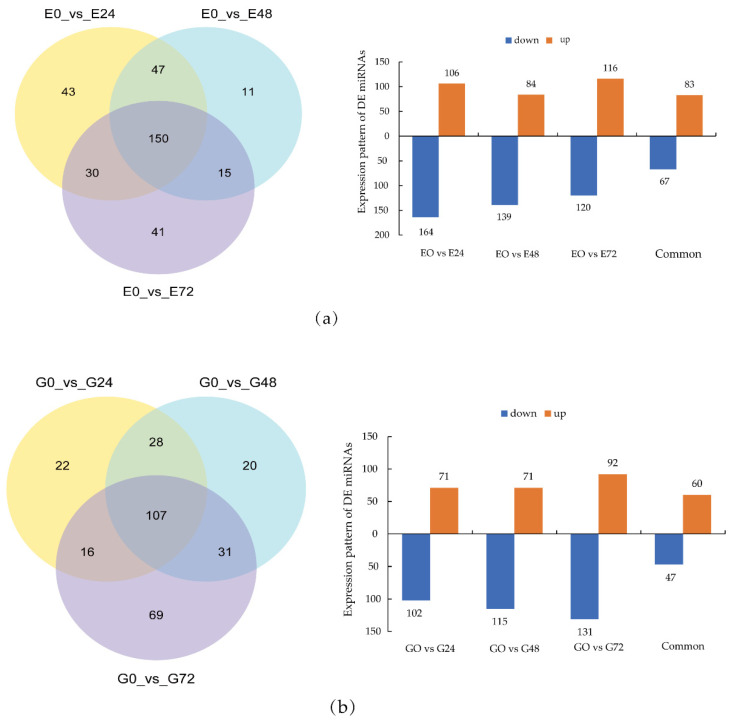
Differentially expressed miRNAs in sweet potato after infection with sweet potato weevils. (**a**) **Left**: changes in miRNA expression in Eshu 6 tubers after infection with sweet potato weevils; **right**: expression pattern of DE miRNAs; (**b**) **left**: changes in miRNA expression in Gaung 87 tubers after infection with sweet potato weevils; **right**: expression pattern of DE miRNAs; (**c**) **left**: common differentially expressed miRNAs in Eshu 6 and Guang 87 tubers under different time treatments; **right**: expression pattern of DE miRNAs.

**Figure 4 genes-13-00981-f004:**
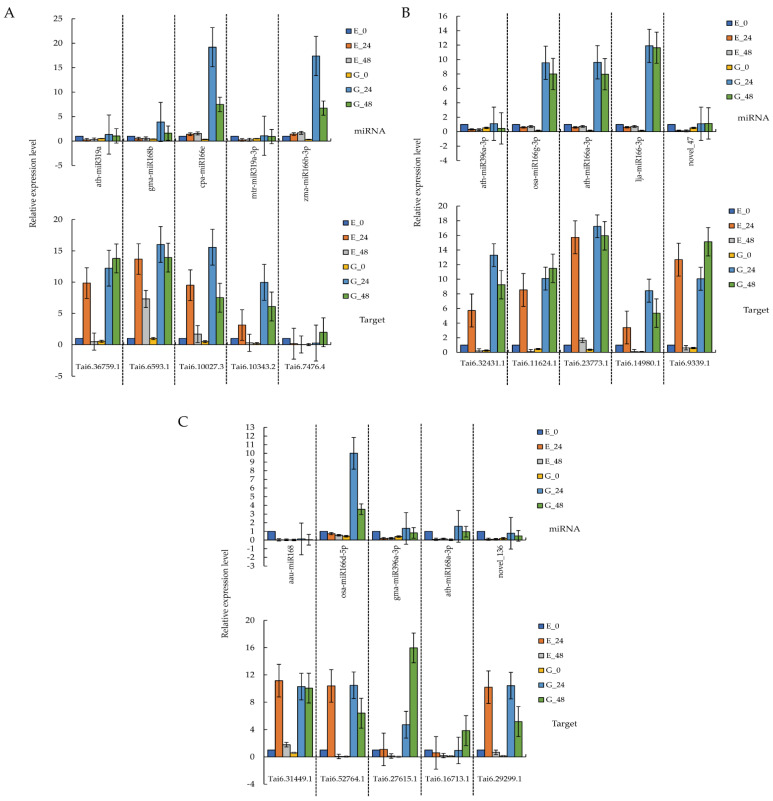
qRT-PCR analysis of miRNAs and target genes. (**A**) qRT-PCR analysis of ath−miR319a, gma−miR168b, cpa−miR166e, mtr−miR319a−3p, zma−miR166h−3p, and their target genes; (**B**) qRT-PCR analysis of ath−miR396a−3p, osa−miR166g−3p, ath−miR166a−3p, lja−miR166−3p, novel_47, and their target genes; (**C**) qRT-PCR analysis of aau−miR168, osa−miR166d−5p, gma−miR396a−3p, ath−miR168a−3p, novel_136, and their target genes.

**Table 1 genes-13-00981-t001:** High-throughput sequencing data statistics for different samples.

Library	Treatment	Sample ID	Total Reads	Clean Reads	Percentage (%)	Mapped Genome	Percentage (%)
Eshu 6	E_0	E_0_1	13,505,817	8,874,325	65.71	7,046,454	66.39
		E_0_2	13,547,525	10,174,405	75.10	4,838,795	50.56
		E_0_3	16,468,157	13,888,078	84.33	6,237,540	46.92
		Mean	14,507,166	10,978,936	75.68	6,040,930	54.62
	E_24	E_24_1	16,990,361	15,477,110	91.09	5,980,088	45.94
		E_24_2	12,898,360	11,156,704	86.50	8,128,053	69.23
		E_24_3	14,544,136	12,618,934	86.76	7,616,186	66.19
		Mean	14,810,952	13,084,249.33	88.34	7,241,442	60.45
	E_48	E_48_1	16,179,390	13,552,999	83.77	9,147,300	68.44
		E_48_2	12,901,334	9,569,488	74.17	10,439,875	57.02
		E_48_3	12,022,565	4,651,802	38.69	12,267,059	71.56
		Mean	13,701,096	9,258,096.333	67.57	10,618,078	65.67
	E_72	E_72_1	15,711,300	13,017,803	82.86	2,910,452	62.57
		E_72_2	14,969,018	13,776,032	92.03	5,994,652	67.55
		E_72_3	16,047,099	13,294,682	82.85	5,389,906	53.80
		Mean	15,575,806	13,362,839	85.79	4,765,003	61.31
Guang 87	G_0	G_0_1	15,031,654	13,592,390	90.43	6,773,030	56.00
		G_0_2	13,460,140	10,613,259	78.85	6,516,875	64.05
		G_0_3	13,994,674	11,699,455	83.60	7,906,581	67.58
		Mean	14,162,156	11,968,368	84.51	7,065,495	62.54
	G_24	G_24_1	17,559,872	17,142,807	97.62	6,482,848	58.11
		G_24_2	14,141,613	12,140,937	85.85	6,207,694	45.06
		G_24_3	13,623,060	11,507,160	84.47	7,465,376	63.02
		Mean	15,108,181.67	13,596,968	90.00	6,718,639	55.40
	G_48	G_48_1	13,554,175	11,845,193	87.39	7,608,730	60.30
		G_48_2	14,180,870	13,364,466	94.24	9,247,207	68.23
		G_48_3	13,115,375	10,018,069	76.38	8,947,533	64.43
		Mean	13,616,807	11,742,576	86.24	8,601,157	64.32
	G_72	G_72_1	12,481,043	11,739,937	94.06	8,511,456	62.62
		G_72_2	13,666,809	12,095,700	88.50	9,168,804	59.24
		G_72_3	19,897,861	18,307,922	92.01	8,574,439	70.62
		Mean	15,348,571	14,047,853	91.53	8,751,566	64.16

Notes: E_0, E_24, E_48, and E_72: the infected Eshu 6 tubers at 0 h, 24 h, 48 h, and 72 h; G_0, G_24, G_48, and G_72: the infected Guang 87 tubers at 0 h, 24 h, 48 h, and 72 h.

**Table 2 genes-13-00981-t002:** Summary of known and novel miRNAs in different samples.

	Mapped Mature Known miRNAs	Mapped Hairpin Known miRNAs	Mapped Mature Novel miRNAs	Mapped Hairpin Novel miRNAs
Total	407	908	298	307
E_0_1	146	475	219	260
E_0_2	161	482	237	265
E_0_3	161	499	243	276
E_24_1	228	577	268	293
E_24_2	205	520	263	288
E_24_3	197	516	266	288
E_48_1	201	510	249	276
E_48_2	211	545	244	277
E_48_3	201	543	207	258
E_72_1	192	490	226	268
E_72_2	202	533	246	277
E_72_3	152	385	205	252
G_0_1	178	500	248	272
G_0_2	153	491	243	269
G_0_3	178	491	245	274
G_24_1	177	514	273	293
G_24_2	239	569	275	294
G_24_3	229	587	274	297
G_48_1	221	566	278	289
G_48_2	221	570	271	290
G_48_3	151	405	214	250
G_72_1	208	578	265	292
G_72_2	192	503	233	267
G_72_3	246	585	269	301

**Table 3 genes-13-00981-t003:** The number of differentially expressed miRNAs in sweet potato after sweet potato weevil infection.

Samples Comparison	Differentially Expressed	Up-Regulated	Down-Regulated
E_0 vs. E_24	270	164	106
E_0 vs. E_48	223	139	84
E_0 vs. E_72	236	120	116
G_0 vs. G_24	173	102	71
G_0 vs. G_48	186	115	71
G_0 vs. G_72	223	131	92
E_24 vs. G_24	131	84	47
E_48 vs. G_48	65	28	37
E_72 vs. G_72	79	48	31

**Table 4 genes-13-00981-t004:** The number of differentially expressed miRNAs in sweet potato after sweet potato weevil infection.

miRNA Family	miRNA	Target_mRNA	Nr
MIR167_1	bna-miR167d	Tai6.4195.1	PREDICTED: glutamate–cysteine ligase, chloroplastic-like [Ipomoea nil]
	bna-miR167d	Tai6.35623.1	PREDICTED: glutamate–cysteine ligase, chloroplastic-like [Ipomoea nil]
	vvi-miR167c	Tai6.4195.1	PREDICTED: glutamate–cysteine ligase, chloroplastic-like [Ipomoea nil]
	vvi-miR167c	Tai6.35623.1	PREDICTED: glutamate–cysteine ligase, chloroplastic-like [Ipomoea nil]
	ptc-miR167f-5p	Tai6.4195.1	PREDICTED: glutamate–cysteine ligase, chloroplastic-like [Ipomoea nil]
	ptc-miR167f-5p	Tai6.35623.1	PREDICTED: glutamate–cysteine ligase, chloroplastic-like [Ipomoea nil]
MIR156	hbr-miR156	Tai6.44728.1	PREDICTED: protein SENESCENCE-ASSOCIATED GENE 21, mitochondrial-like [Ipomoea nil]
n_MIR318	novel_318	Tai6.52197.1	PREDICTED: ethylene receptor 1 isoform X1 [Ipomoea nil]

## Data Availability

Not applicable.

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
