# Peer review of "Identification of miRNAs in Response to Sweet Potato Weevil (Cylas formicarius) Infection by sRNA Sequencing"

_genes, 2022, doi:10.3390/genes13060981_

Round 1

Reviewer 1 Report

As an introductory study, I didn't expect any definitive results. Congratulations on identifying a number of novel miRNA.  I recommend the authors not overstate the significance of this work.  Some of the miRNAs could prove to play a significant role in plant-insect interactions but it doesn't seem to me that you've proven any such thing here.  With a little work, this paper could be of use to researchers studying the role of miRNAs in sweet potato.  

Some specific comments:

  1. In the abstract, line 24, authors suggest they treated tubers for different lengths of time but do not make it clear but they used to treat the tubers. Line 33 indicates the treatment is sweet potato weevil infection but this should be made clearer sooner in the abstract, before the miRNA findings are mentioned.
  2. Explain what is meant by the tobacco-TBM interaction mentioned on lines 45/46 is if relevant to your paper.
  3. On lines 70-71, you suggest the “key” miRNAs have not been identified. How do you know this?
  4. The authors appear to be relying on a transcriptome library for the sweet potato (Sections 2.5 and 2.6). The origin of the transcriptome and its accessibility to researchers should be mentioned. For example, is the transcriptome part of a previously published study?  Did the authors rely on the genome for the tuber mentioned in Line 140?  If so, then the genome sequence should be mentioned earlier.
  5. Line 113 “chage” = “change”
  6. The number of reads obtained for each library are perhaps best left in the table. Mentioning them in the text isn’t helpful.
  7. Lines 246/247 the authors suggest the genes targeted by certain miRNAs identified in their study (Table 4) play key regulatory roles in insect resistance. The observation that the levels of these miRNAs change in the tubers as a function of exposure to weevils doesn’t seem adequate for concluding exactly what the miRNAs do.
  8. I’m curious why the “key” miRNAs shown in Table 4 were not included in the qRT-PCR study. Why were these particular 15 miRNAs chosen for study? It may be that these 15 were related to miRNAs identified in other studies as being of significance. If so, state that.
  9. Line 458 -infestation is misspelled.

Author Response

Dear Reviewer,

We would like to thank you for your careful reading, helpful comments, and constructive suggestions, which has significantly improved the presentation of our manuscript. We have carefully considered all comments from the reviewers and revised our manuscript accordingly. The manuscript has also been double-checked, and the typos and grammar errors we found have been corrected. In the following section, we summarize our responses to each comment from the reviewers. We believe that our responses have well addressed all concerns from the reviewers. We hope our revised manuscript can be accepted for publication.

Reviewer 2 Report

Lei et al identified miRNA profiles in the sweet potato infected by sweetpotato weevil by using sRNA profiling. Authors have shown a new approach to identifying the regulatory role of miRNA in the response to infections. I believe this is one of the first reports of this study on the miRNA profiling in sweetpotato weevil infection in sweet potato. In addition, the findings of these studies have very good potential for the target selections in the application of insect resistance and further studies in the molecular mechanism. Although more details study of some specific pathways selected based on the expression profiling could be discussed more, the current form still gives reader new information that can be utilized in future research. 

Overall, the authors have provided the study's scientifically sound approach with a concise discussion, in addition to the results and supplementary files provided.

Author Response

Dear Reviewer,

We are very grateful to Reviewer for reviewing the paper so carefully. We have tried our best to improve and made some changes in the manuscript. We thank the reviewer for reading our paper carefully and giving the above positive comments.

With kindest regards.